# Bacteriophages and Food Production: Biocontrol and Bio-Preservation Options for Food Safety

**DOI:** 10.3390/antibiotics11101324

**Published:** 2022-09-28

**Authors:** Mary Garvey

**Affiliations:** 1Department of Life Science, Atlantic Technological University, F91 YW50 Sligo, Ireland; mary.garvey@atu.ie; 2Centre for Precision Engineering, Materials and Manufacturing Research (PEM), Atlantic Technological University, F91 YW50 Sligo, Ireland

**Keywords:** bacteriophage, biocontrol, disease mitigation, food sustainability, potency, endolysins

## Abstract

Food safety and sustainable food production is an important part of the Sustainable Development goals aiming to safeguard the health and wellbeing of humans, animals and the environment. Foodborne illness is a major cause of morbidity and mortality, particularly as the global crisis of antimicrobial resistance proliferates. In order to actively move towards sustainable food production, it is imperative that green biocontrol options are implemented to prevent and mitigate infectious disease in food production. Replacing current chemical pesticides, antimicrobials and disinfectants with green, organic options such as biopesticides is a step towards a sustainable future. Bacteriophages, virus which infect and kill bacteria are an area of great potential as biocontrol agents in agriculture and aquaculture. Lytic bacteriophages offer many advantages over traditional chemical-based solutions to control microbiological contamination in the food industry. The innate specificity for target bacterial species, their natural presence in the environment and biocompatibility with animal and humans means phages are a practical biocontrol candidate at all stages of food production, from farm-to-fork. Phages have demonstrated efficacy as bio-sanitisation and bio-preservation agents against many foodborne pathogens, with activity against biofilm communities also evident. Additionally, phages have long been recognised for their potential as therapeutics, prophylactically and metaphylactically. Further investigation is warranted however, to overcome their limitations such as formulation and stability issues, phage resistance mechanisms and transmission of bacterial virulence factors.

## 1. Introduction

There is increasing demand for high quality food produce to meet the needs of an ever-increasing human population. Sustainability issues are evident where increasing food production impacts negatively on the planet, biodiversity and animal health which are major concerns of the One Health agenda. The United Nations (UN) have listed zero hunger incorporating food safety and food security and climate action as their Sustainable Development Goals (SDGs) where action must be taken to ensure sustainable processes are implemented. Indeed, sustainable agriculture may be considered the most significant part of the SDGs, and it directly or indirectly encompasses the 17 SDGs [1]. Non-sustainable food production methods are a major contributor to greenhouse gas emissions (GHG), water pollution, loss of biodiversity and deforestation at a global scale [2] while millions of persons still suffer from hunger and malnourishment on an ongoing basis. Indeed, the UN and Food and Agriculture Organization (FAO) have outlined the ongoing food crisis, where 828 million people were affected by hunger in 2021 an alarming increase from 150 million in 2019 [3]. Agri-food production currently uses approximately 34% of global land area which promotes industrial land use, excessively reducing biodiversity while further increasing nitrogen aquatic pollution [4]. Livestock production results in approximately half of the GHG emissions (12–18% of global emission’s) resulting from food production, where cattle ranching also leads to deforestation and water pollution [5]. Aquaculture has many negative impacts on aquatic environments such as water pollution, transmission of disease to wild fish, loss of wild fish species, algae blooms and eutrophication [6]. Infectious disease in agriculture, aquaculture and livestock food production is another food sustainability issue, where infectious disease results in crop losses, animal welfare issues, food wastage, water, and soil pollution with antimicrobial agents [7]. Food production processes are also known to directly impact on emerging infectious disease in humans and animals which also contributes to zoonotic transmission [8]. Additionally, food microbial contamination results in disease outbreaks, morbidity and mortality globally resulting in food safety issues which are difficult to monitor, predict and survey. Coupled with the threat of antimicrobial resistance (AMR), foodborne infectious disease will undoubtedly become a greater economic burden and public health risk. Multidrug resistant (MDR) bacterial species have been isolated from food types (meat, poultry) including ampicillin, tetracycline, quinolone and sulfonamide resistant *Escherichia coli* and *Salmonella* spp. Extended spectrum beta-lactamase (ESBL) and carbapenemase producing *Salmonella* and *E. coli* was also identified in poultry [9]. Carbapenem resistant and ESBL producing Enterobacteriaceae including *E. coli* are currently listed as critically important on the World Health Organization (WHO) priority pathogen list with fluoroquinolone resistant *Salmonella* listed as highly important. Controlling infectious disease in food producing animals and crops is effective in preventing human disease [10], food spoilage and food waste thereby contributing to food sustainability. Biocontrol measures, however, are currently based on antimicrobial agents including disinfectants, antibiotics, antifungals, pesticides (fungicides) and are not considered environmentally safe or sustainable green methods. Food preservation methods may also negatively impact on the organoleptic quality and nutritional status of food. Terrestrial and aquatic pollution with such compounds and excessive nutrients also puts pressure on organisms and ecosystems. Additionally, pesticide use is under regulatory restrictions with a move towards greener alternative agrochemicals being promoted [1]. Consequently, there is an urgent need to implement alternative safer options such as biopesticides. Biopesticides can be defined as microbial species or naturally occurring compounds which inhibit the growth of crop pests thereby safeguarding crops pre harvest. Many bacterial and fungal species can act as biopesticides with *Bacillus thuringiensis* currently occupying 90% of the market [11]. Bacteriophages (phages), viruses that infect bacterial species also offer potential as biopesticides or biocontrol agents in food production at pre- and post-harvest. This review outlines the application of phages as biocontrol agents in food production increasing food safety while reducing the environmental impact of food production systems.

## 2. Bacteriophages—Evolution as Antimicrobials

Bacteriophages out number bacterial cells by 10-fold, present in the biosphere and intestines of animal and human species, making them the most abundant species on earth [12]. The phage genome ranges in size (3.4 kb to ca. 500 kb) having no single gene are present in all phage genomes meaning that phages contain a large range of genes and proteins currently not investigated [13]. Phages have a lytic or lysogenic life cycle where the former results in host cell lysis and death and the latter results in viral genome insertion in to host genomic material [14] post infection. Phages attached to the host cell via varied host receptors (proteins, carbohydrates, lipopolysaccharides) having a narrow host rage and increased specificity [15]. Lysogenic phages insert their genome into the hosts bacteria chromosome termed prophage where many prophages can be present (poly-lysogenic strains) in one bacterial genome [16]. Lytic phages play an essential role in the biosphere modulating evolutionary and ecological activity in microbial communities, controlling population size, stimulating biomass turnover releasing nutrients where lysogenic species can alter bacterial genomes and evolutionary processes [17]. The antibacterial activity of phage’s had long been considered for the treatment of infectious disease in the pre antibiotic era. It was the microbiologists Felix d’Herelle that first introduced the idea of phages in clinical medicine as prophylactic and metaphylactic treatment and was recognised with the discovery of phages in 1917 [18]. d’Herelle recognised the selectivity and potency of phages for the cellular destruction of disease-causing bacteria while being biocompatible with the host patient. d’Herelle conducted many trials using phage’s for intravenous (IV) administration against invasive bacterial infections after successfully treating chickens infected with *Salmonella gallinarum* [19]. Phage therapy against bacillary dysentery was successful in 1921 using phage’s capable of infecting *Shigella dysenteriae* [20]. Research in treating cholera in India also proved promising with a significant decrease in mortality (62.8% to 8.1%) following phage administration, where phage’s specific for the cholera pathogen were also added to the drinking water of villages preventing outbreaks [19]. Phage therapy, however, became overlooked with the advance in antibiotic discovery and development from the 1930s (sulpha drugs) and 1940s (penicillin) [21]. Issues with phage research included inconsistent results, issues with reproducibility, administration doses and limited availability of genetic information [20] additional issues relating to large scale production, formulation, stability, and storage must also be considered [14].

While the United States and Western Europe abandoned phage studies, Eastern Europe namely Poland continued phage research which has demonstrated efficacy against AMR pathogens. Studies by Smith and Huggins in the 1980s demonstrated the activity of phage R against K1 *E. coli* in mice where one dose of phage R was more effective than 8 doses of the antibiotic streptomycin via intramuscular (IM) administration where phages were not detected in the muscle, spleen, blood and liver of dosed animals at 16 h post administration [19] demonstrating phage clearance. Indeed, enterally, and parenterally administered phage’s have demonstrated efficacy for approximately 90 years in Eastern European studies in the absence of patient side effects [20]. With the evident and alarming threat of AMR, phage therapy may now offer much needed options where current therapeutic methods are failing. For example, the development of phage therapy (Table 1) may offer therapeutic options for the treatment of Clostridioides difficile infection (CDI) (formerly *Clostridium difficile*), an MDR bacterial infection associated with high rates of mortality [22]. At present, 6 treatment clinics have emerged globally (the US, the UK, the Republic of Georgia, Poland and Belgium) offering phage therapy for alleviating infectious disease [20].

## 3. Bacteriophages and Food Production

The WHO states that 600 million cases of foodborne illness occur yearly with 420,000 deaths globally with a prevalence of 30% to 40% among children < 5 years [23]. The Foodborne Disease Burden Epidemiology Reference Group (FERG) established by the WHO monitors the 31 foodborne pathogens associated with severe morbidity and mortality in humans where the most prominent species causing disease are *E. coli*, *Campylobacter* sp., non-typhoid *Salmonella enterica* and *Shigella* species [23]. Food safety is an important topic for public health and economic strategies as it results in morbidity, mortality, clinical costs and industrial economic burdens. There are many routes of transmission in the epidemiology of foodborne infectious disease, where animals can be asymptomatic carriers of pathogens such as *Salmonella*, *E. coli*, *Campylobacter* and *Listeria* but can transmit these species to crops, slaughter facilities, and directly to humans. Additionally, food production facilities offer ideal conditions for the formation of bacterial biofilms which can harbour and spread foodborne pathogens [24].

Control measures are implemented to prevent and control microbial contamination including Good Manufacturing Practices (GMP) and Hazard Analysis Control Points (HACCPs) in food production and preparation industries [25]. Food preservation tactics, disinfection with antimicrobial biocides and antimicrobial storage protocols are stringently monitored. According to the European Food Safety Authority (EFSA) report of 2020 however, *Campylobacter* resulted in 317 foodborne outbreaks reported by 17 EU member states with 1319 cases of illness and 112 hospitalisations [26]. Additionally, there was 3686 cases of foodborne Salmonellosis and 694 outbreaks with *Listeria monocytogenes* causing 16 foodborne outbreaks in 7 EU member states with 120 cases of illness, 83 hospitalisations and 17 deaths. Shiga toxin producing *E. coli* (STEC) as a foodborne pathogen resulted in 34 outbreaks, 208 cases, 30 hospitalisations and 1 death reported in 2020 [26]. The emergence of AMR in pathogenic species is also coupled with biocidal resistance which means disinfection regimes have become less reliable particularly for Gram-negative species [27]. Foods are recognised as important reservoirs of many Gram-negative species including *E. coli*, *Klebsiella* spp., and *Acinetobacter* spp. [28]. Disinfection commonly in use in food production include quaternary ammonia compounds (QACs) dodecyldimethylammonium chloride (DDAC) and benzalkonium chloride (BAC), hypochlorite’s, iodophors and chlorine dioxide-based solutions [29]. Resistance to QACs and BAC has emerged in MDR species of *Pseudomonas* and *Enterobacteriaceae* due to the presence of qac genes [27]. Indeed, many of the WHO high priority “ESKAPE” pathogens (*Enterococcus* spp., *Staphylococcus aureus, Klebsiella pneumoniae, Acinetobacter baumannii, Pseudomonas aeruginosa,* and *Enterobacter* spp.) associated with foodborne transmission are also multidrug-resistant (MDR), extensively drug-resistant (XDR), and pan drug-resistant (PDR) species also displaying biocidal resistance [27]. Methicillin resistant *Staphylococcus aureus* (MRSA) is also a common foodborne pathogen. Furthermore, issues have arisen with the presence of biocidal residues in food products exceeding the maximal residual limits (MRLs) [29]. Such biocides are common pollutants in waste waters and are toxic to aquatic organisms [25]. As efforts are made to combat biocidal resistance this issue will undoubtedly proliferate, therefore, there is an urgent need for safer, greener alternatives to act as biocontrol agents in food production.

### 3.1. Bacteriophages as Potential Biocontrol Agents for Food Safety

The use of bacteriophages is increasingly recognised as a green biocontrol technology with high specificity at targeting bacterial pathogens present in many settings including food production [23]. Lytic phages infect and kill target bacterial species, are self-replicating while remaining biocompatible with animal and human species without causing environmental issues making them ideal candidates for food biocontrol and bio-preservation agents (Table 2) [14]. Phages as biocontrol agents preventing infectious disease in food producing animals and crops and subsequent foodborne zoonosis is an area of much interest. For livestock application, phages can be administered via animal feed or sprayed on animal bodies prior to sacrifice/slaughter to prevent meat contamination at harvest. Post-harvest application, phages may offer disinfection of surfaces via spraying with phage enzymes potentially acting as food preservatives [24].

#### 3.1.1. Livestock Food Production

Phage’s have long been recognised as problematic contaminants of dairy food production where they negatively impact fermentation processes as they lyse the fermentation bacterial species. Raw milk often contains phage concentrations exceeding 10^4^ PFU per ml with concentrations of 10^6^–10^7^ plaque-forming units (PFU) per mL resulting in fermentation failures [39,40]. The presence of phages and prophages in lactic acid bacteria is widely recognised with *Streptococcal* phages also detected in milk samples used for yogurt production [41]. The presence of these phages in starter cultures which contain lactic acid bacteria typically strains of *Lactococcus lactis, Streptococcus thermophilus, Leuconostoc* sp., and/or *Lactobacillus* sp. negatively impacts the production of fermented products [42] inflicting economic losses on dairy food production companies. Phages are typically heat resistant allowing them to resist pasteurisation and temperatures applied in food manufacturing processes but may be controlled via UV disinfection in dairy parlours [16]. Bacteriophages however, also offer potential prophylactic and metaphylactic disease control strategies at farm level for the treatment of infectious disease in animal herds. Clinical and subclinical mastitis, lameness, respiratory disease and metritis are infectious disease impacting livestock production, leading to reduced productivity, increased animal culling, veterinary costs and economic burden on farmers [10]. Phage K for example has been studied for treating *S. aureus* subclinical mastitis in dairy cattle with a phage cocktail of ΦH5 and ΦA72 inhibiting *S. aureus* growth in treated milk [43]. The T4-like virus vB_EcoM-UFV13 phage may offer treatment options for environmental mastitis caused by *E. coli* where successful therapy was achieved in mice models [44]. Studies on clinical and subclinical cases of mastitis are hindered however, by the milk matrix present in the udder. Indeed, the T4-like phage may also offer biocontrol options for the highly fatal calf diarrhoea caused by enterotoxigenic *E. coli* (ETEC) [25].

The double stranded DNA mycobacter”opha’e D29 may offer some biocontrol options against several species of mycobacteria, including the pathogenic *Mycobacterium tuberculosis* (TB) [45]. *Salmonella* and *Campylobacter* are a bacterial species causing infectious disease and food production issues in poultry farming. The use of phages to control *Salmonella* in chickens and pigs has proven effective against *S. enteritidis, S. hadar* and *S. typhimurium* [46]. *E. coli* induced diarrhoea in pigs’ results in high mortality and morbidity, decreased growth rate, and significant economic losses. Studies have shown a reduced presence of intestinal *E. coli* in phage treated pigs compared to untreated control groups [47]. The *Campylobacter* phage CP220 demonstrated an ability to reduce *C. jejuni* in chickens by ca. 2 log after single dosing [48]. Additionally, respiratory tract infection colibacillosis, caused by avian pathogenic *Escherichia coli* (APEC) is responsible for high mortality rates in poultry farming. Studies assessing combination of bacteriophages SPR02 and DAF6 to treat chicken colibacillosis via aerosol and IM administration demonstrated a > 80% survival rate via IM route [43]. When combined with 50 ppm enrofloxacin administered via water the survival rate was 100% for infected chickens [49].

Phage bio-preservation products have been tested on foodborne pathogens including *Shigella, Staphylococcus, L. monocytogenes, E. coli* and *Salmonella* species [32]. The phage cocktail product ListShield^TM^ was approved in 2006 by the FDA as a food preservative against 170 strains of *L. monocytogenes* leading the way for the development of many commercial products (Table 1). ListShield^TM^ provided a 95% reduction in *L. monocytogenes* viability when applied at 1 mL per 500 cm^2^ of food produce before packaging [50]. In 2007 the FDA approved the phage product Listex P100 as a bio-preservative in ready to eat (RTE) meat food products [37] which is currently classed as General Recognised as Safe (GRAS) for use in food production [24]. EcoShield^TM^ approved by the FDA in 2011 for use on red meat against *E. coli* namely *E. coli O157:H7* was developed by Intralytix, USA, followed by SalmoFresh^TM^ against *Salmonella enterica* for the biocontrol of fruit, vegetables, and seafood [20]. EcoShield^TM^ contains 3 lytic phages against *E. coli O157:H7* and achieved a 95% loss of bacterial viability at 10^6^ PFU per gram via spraying within 5 min [50]. ShigaShield^TM^ also by Intralytix, has demonstrated efficacy against *Shigella* species in smoked salmon and yogurt in a concentration dependent manner and is currently under review as a GRAS product [51]. Indeed, the FDA has approved several phages or phage cocktails for commercial use via spray or dip application with lab studies also confirming their efficacy on food products encouraging industrial applications [39]. As bio-preservation agents, *Listeria* Siphoviridae phages LMP1 and LMP7 and FWLLm1 and FWLLm3 when combined with a bacteriocin inhibited the growth of *L. monocytogenes* at refrigeration temperatures [41]. Bacteriophages PA13076 and PC2184 at 1 × 10^8^ PFU/mL also proved effective at inhibiting *S. enteritidis* in pasteurised whole milk at refrigeration temperature providing ca. 4 log loss of bacterial viability [52]. Similarly, phage LPSE1 demonstrated the ability to decontaminate milk contaminated with *S. enteritidis* at 28 °C achieving a ca. 2.37 log bacterial death [25].

#### 3.1.2. Agricultural Food Production

Plant pathogens (phytopathogens) affecting food crops result in plant disease, crop losses, food shortages and significant economic losses globally. Bacterial pathogens associated with agricultural infectious disease include *Pseudomonas* spp., *Xanthomonas* spp., *Erwinia* spp., *Ralstonia* spp., *Agrobacterium* spp., *Xylella* spp., *Pectobacterium* spp., and *Dickeya* spp. [53]. Currently, antibiotics primarily streptomycin, kasugamycin and tetracyclines and antimicrobial pesticides are used to prevent crop and plant diseases resulting in environmental pollution and proliferation of AMR. Copper is currently the antimicrobial of choice for crops; however, its application negatively impacts pollinating insects, results in phytotoxicity, and bioaccumulates in soil and water reducing biodiversity. In 2005 the FDA approved the bacteriophage-based product Agriphage™, by OmniLytics Inc. for the treatment of bacterial spot disease on crops [35]. Phages vB_DsoM_LIMEstone1, vB_DsoM_LIMEstone2 amongst others have demonstrated efficacy at controlling the causative agents of soft rot in potatoe crops [53]. Phages or phage cocktails have also demonstrated efficacy against *Sarcoptes scabies* on raddish crops, *Xanthomonas axonopodis* on onion leaf, Pectobacterium *carotovorum* infection of lettuce, blight in leeks and rice and bacterial spot and wilt in tomatoes [53]. Furthermore, phage therapy reduced fungal infection by *Psuedomonas tolaasii* in mushroom crops [54].

The environmentally present pathogenic species *Ralstonia solanacearum* is the causative agent of bacterial welt in many crops, and is therefore, a pest of economic and environmental importance in the EU [55]. The studies of on 3 isolated phages named vRsoP-WF2, vRsoP-WM2, and vRsoP-WR2, against *R. solanacearum* in environmental water achieved excellent bacterial death of 5 log cfu in less than 10 h at 24 °C [56]. The Gram-negative bacterial genus *Xanthomonas* which contains 27 species being pathogenic to ca. 400 plant species including sugar cane, beans, cassava, cabbage, banana, citrus, tomatoes, pepper and rice is currently managed with copper-based pesticides, antibiotics and plant culling [57]. Two *Xanthomonas* phage products manufactured by AgriPhage have demonstrated efficacy against spoilage of tomato and pepper crops and citrus canker disease [58]. Phage biocontrol and plant protection particularly for epiphytic microbes can be impacted by environmental conditions such as sun light and UV rays, temperature fluctuations, rain and additional crop pesticides [35] therefore, application considerations must be addressed to increase the presence of phage on plant surfaces. Factors to consider when applying phage biocontrol solution to crops include frequency, contact time, UV exposure, the epiphytic or endophytic nature of the pathogen and temperature. Phage biocontrol in agriculture therefore, should be considered under an integrated plant protection (IPP) protocol to predict, prevent, detect, treat and monitor bacterial disease in crops [59]. Phage based detection assays for human and animal pathogens have been developed with potential for use in crop and plant pathogens, for example a phage based quantitative PCR assay for *R. solanacearum* has shown promise [60].

#### 3.1.3. Aquaculture

Aquaculture and fish food production is a growing food sector globally. According to the FAO global fish production reached 179 million tons in 2018 with a 500% increase in fish food production in the past 30 years [61]. Fish and seafood are also prone to infectious disease, spoilage and contamination where antibiotic treatment is currently in use prophylactically and metaphylactically. Due to the ease of transmission of pathogens in aquatic environments between wild and farmed fish control and treatment options for a vast range of fish pathogens including vaccines and antibiotics are often insufficient, ineffective, or unfeasible [62]. Vibriosis, Aeromonasis, Edwardsiellosis, Mycobacteriosis, hemorrhagic septicemia, ulcer disease, and *Flavobacterium*, are the main bacterial disease of fish industries [63]. *Vibrio* species including *V. harveyi* which frequently display MDR are responsible for disease outbreaks which can result in 98.5–100% of mortality fish hatcheries and shrimp [64].

The use of phages as biocontrol agents in aquaculture has demonstrated efficacy via direct application in water, oral administration via food and injection [63]. For example, the Phage VPp1 and A3S and Vpms1 successfully controlled *V. parahaemolyticus* in oysters and shrimp, respectively, [65]. Studies have demonstrated the efficacy of phage PLgY-16 when administered orally or intraperitoneally against lactococcosis in yellowtail (*Seriola quinqueradiata*) infected with *L. garvieae* and the use of phage PPpW-4, via fish feed, to combat the bacterial hemorrhagic ascites disease in ayu fish (Plecoglossus altivelis) caused by *P. plecoglossicida* [53]. *Flavobacterium psychrophilum*, a Gram-negative bacterium is the causative agent of bacterial cold-water disease (BCWD) a fatal infectious disease of several farmed salmonid species [66] displaying MDR and biocide resistance [66]. Studies demonstrate the efficacy of phage PSV-D22 and similar phages for the treatment of *F. psychrophilum* present in live fish eggs or fry [66]. Aquaculture infectious diseases caused the genus *Vibrio* have demonstrated susceptibility to phage biocontrol. The application of phages in the treatment of Penaeus monodon larvae infected with *V. harveyi* resulted in an 85% survival of the larvae compared to 65–68% survival following antibiotic treatment [53]. Proteon Pharmaceuticals has developed a commercial phage product, BAFADORR, to eliminate *Pseudomonas* and *Aeromonas* infections in aquaculture [14].

The application of phages in aquaculture raises some challenges in terms of route of administration and environmental factors. Water pH, salinity, temperature, organic load and UV exposure and phage contact to the farmed species may impact phage activity in an aquatic setting [64]. Ensuring phage contact with the causative agent of disease is paramount to treatment therefore, the route of administration must be considered. Efforts have been made by researchers to increase phage exposure by way of bacteriophage-based edible antimicrobial coatings on fish feed which improved the stability of phages on fish feed pellets and survival during feed storage [35]. Studies showed that phage specific for *E. coli* and *Vibrio* spp. within the coatings produced greater bacterial death in vitro [35]. Interesting studies by Gabiatti et al., (2018) describe the use of *Bacillus* endospores to house and protect the phage genome under harsh storage conditions with viral propagation occurring with bacterial germination [67]. This method may aid in phage use as biocontrol agents by allowing for long term storage particularly if non-pathogenic spore forming bacterial species are used as hosts.

### 3.2. In the Control of Bacterial Biofilms

Biofilms are microbial communities which grow attached to surfaces, where cells are surrounded by a protective layer of lipids and an extracellular polysaccharide (EPS) matrix. Bacterial biofilms are abundant on moist surfaces, biotic and abiotic surfaces, and in the phyllosphere of plants where a suitable environment for bacterial survival and proliferation is established as protection against harsh conditions [58]. The biofilm is an important microbial virulence factor for plant, animal and human pathogens allowing for AMR, resistance to host immunity, and attachment to plant vascular tissue. Additionally, the formation and growth of biofilms on food production surfaces and equipment is a major issue for food production where biocidal resistance is also present [29]. Foodborne pathogens forming resistant biofilm communities include *L. monocytogenes, Salmonella, E. coli, Yersinia* which can also colonise plant tissues as biofilm communities [35]. Biofilms of the important plant pathogens *Xanthomonas* species [58] and *Ralstonia solanacearum* allows for their endophytic colonisation of many crop plants [68]. The most common bacterial pathogens forming biofilms relevant to aquaculture include *Vibrio* spp., *Aeromonas hydrophila, Salmonella* spp., and *L. monocytogenes. Vibrio parahaemolyticus* can form biofilms on the chitin of oysters where *Vibrio cholerae* can form biofilms on phytoplankton and zooplankton [25]. Studies have demonstrated effective eradication of biofilms formed by pathogens *Streptococcus species*, *E. coli*, *Pseudomonas aeruginosa*, *Staphylococcus aureus*, and *Enterococcus faecalis* relevant to dairy food and fresh food production. Phage EFDG1 (orthocluster II) for example has proven effective at eradicating two-week-old biofilms of *E. faecalis* V583 [37]. *L. monocytogenes* biofilms also appear sensitive to phage biocontrol where PhageGuard Listex using phage P100 was effective at eradicating biofilms on stainless steel surfaces [69].

The use of phage cocktails may offer antibiofilm activity against a mixed biofilm of *Staphylococcus aureus* and *Pseudomonas aeruginosa* [70]. *Staphylococcal* phage K and a mixture of derivative phages with broader host range prevented *S. aureus* biofilm formation over incubation periods of 48 h to 72 h at 37 °C [25]. The phages CP8 and CP30 prevented the formation of *C. jejuni* biofilms (poultry pathogen) in vitro by ca. 3 log cfu/cm^2^ [35]. Biofilms of the plant pathogen *Xanthomonas* appear sensitive to the phage X3 and XacF1 via spray application on rice plants preventing disease symptoms [57]. The use of phages against biofilms of *Flavobacterium psychrophilum* in aquaculture shows promise as certain phages and phage combinations demonstrated an ability to inhibit biofilm formation and biomass reducing properties [66]. Phage cocktail preparations have shown good efficacy at preventing and eradicating biofilm communities [71]. The efficacy of phages or phage cocktails against biofilms structures is influenced by many factors including intrinsic structure or composition of the food item, species present, phage type, phage enzyme production and biofilm penetration. Studies have shown that genetically modifying phages for increased enzyme production and biofilm penetration increases phage activity. Genetically modified (GM) T7 *E. coli* phage for example expresses intracellular hydrolase which is released into the EPS matrix promoting biofilm degradation achieving a 99% elimination of biofilms [72,73]. Phages may also encode extracellular polysaccharides (EPS) depolymerases, enzymes allowing for increased biofilm penetration [70]. Indeed, phage depolymerases can degrade the cell-associated polysaccharides and the EPS, to aid phage adhesion to biofilm cells [35]. The bacterial cells in the deeper biofilm layers where oxygen and nutrition supply are limited have reduced growth rates however, this phage replication is also hindered, therefore, in these layers phage resistance may be present [71]. Phages can be used in combination with other antibacterial agents including antibiotics, bacteriocins and disinfectants, to improve the effectiveness of biofilm elimination.

### 3.3. Phage Enzymes as Biocontrol Agents

Bacteriophages code for an enzyme family termed the peptidoglycan cell wall hydrolases divided into the virion-associated peptidoglycan hydrolases (VAPGHs) and the endolysins [25]. The VAPGHs function at the beginning of the phage life cycle by forming a small hole in the cell envelope to allow for the insertion of phage genetic material into the host cell via the tail tube [74] and subsequent phage infection. Most hydrolases are in the group of O-glycosyl hydrolases utilising a water molecule to cleave the O-glycosidic bonds of the polysaccharide. This group includes sialidases, rhamnosidases, levanases, xylanases, and dextranases [75]. Hydrolases frequently studied include sialidases or endo-N-acetylneuraminidases originating from *Escherichia coli* K1 specific bacteriophages such as K1A, K1E, K1F, K1–5, 63D, CUS-3, Φ1.2, and Φ92 [75]. Endolysins degrade the bacterial cell wall at the end of their lytic cycle by cleaving peptidoglycan resulting in cell wall lysis and the release of the new phages [76]. They work in conjunction with holin proteins which penetrate the cytoplasmic membrane making holes enabling the endolysins to interact with the peptidoglycan [77]. There are 5 types of endolysins (termed enzybiotics) based on the peptidoglycan bonds which they cleave, glucosaminidases, lytic transglycosylases, muramidases, amidases, and endopeptidases [35].

Endolysins have activity against Gram-positive species but are hindered by the outer membrane of Gram-negative species. As such, they can be applied exogenously to Gram-positive bacteria as they are absent an outer membrane, where additional factors must be added to destabilise the outer membrane in Gram-negative species [74]. Endolysin LysH5 for example, produced by the *S. aureus* phage vB_SauS-phiIPLA88, provided a ca. 3 log death of *Staphylococcus* biofilms where LysCSA13 from the CSA13 phage also provided a 90% removal of *Staphylococcus* biofilms compared to the untreated control. A combination of the endolysin LysK and DA7 depolymerase also demonstrated efficacy against *Staphylococcus* biofilms [35]. Importantly, the endolysin Trx-SA1 (20 mg/day) appeared effective at alleviating mild clinical mastitis resultant from *S. aureus* infection in bovine therapeutic trials [25]. The Ply6A3 endolysin demonstrated efficacy against by *A. baumannii* sepsis in a mouse model via intraperitoneal inoculation with a 70% survival rate of infected animals with no negative impacts on the animal [76]. Endolysins have reported activity against the robust Gram-positive pathogen *C. perfringens* [46]. Endolysins have also proven effective against MDR species where TSPphg is active against MDR strains of MRSA [76]. The endolysin ABgp46 from *Streptococcus pyogenes* proved effective at treating MDR species of as *P.* aeruginosa, *A. baumannii*, and *Salmonella typhimurium* causing bacteraemia in mice [43].

Chelators including ethylenediaminetetraacetic acid (EDTA) and organic acids (citric and malic acids) are often used as outer membrane destabilisers for Gram-negative species [77]. Biofilms of the Gram-negative *S. enterica* serovar *Typhimurium* were treated with endolysin Lys68 achieving 1 log reduction in viable biofilm cells after 2 h of incubation in the presence of outer membrane permeabilizers [25]. Phage depolymerase proved effective at removing a biofilm of *Klebsiella* sp. after 4 h on food production surfaces (80% loss) which increased in combination with chlorine dioxide (92% loss) [78]. The phage Φ6 producing the P5 protein has demonstrated efficacy under membrane destabilising condition against *Pseudomonas* species, *E. coli, Salmonella typhimurium* and *Proteus vulgaris* where P5 acts as a VAPGH and an endolysin [74]. The *Salmonella* endolysin Lys68 proved effective at inhibiting Gram-negative species *Salmonella, Acinetobacter, Pseudomonas, Shigella, E. coli O157:H7, Cronobacter sakazakii*, and *Proteus* when combined with citric or malic acid [77]. Certain endolysins have demonstrated activity against Gram-negative bacteria in the absence of a destabiliser, the *Salmonella* phage SPN9CC endolysin was effective *E. coli* and the *A. baumannii* endolysin LysAB2 demonstrated activity against *Acinetobacter* and *E. coli*, however at a reduced level [79]. Furthermore, endolysins demonstrate an ability to interact with bacterial spores as well ss their vegetative counterparts [57].

Studies describe the endolysin biocontrol of foodborne pathogens including Gram-positive *Streptococcus pneumonia, S. aureus, L. monocytogenes, Enterococcus faecalis*, and *Clostridium perfringens* [77]. As food biocontrol option endolysin can be applied to food products directly as a bio-preservative or in combination with current preservation methods. For example, an endolysin from the bacteriophage ΦH5 specific for *Staphylococcus* species completely reduced *S. aureus* microbial load in pasteurised milk in 4 h and by 1 log in 60 min [74]. The endolysin LysZ5 which is specific for *Listeria* sp. inhibited *L. monocytogenes* by ca. 4 log after 3 h incubation in soya milk [77]. The endolysin of PlyP100 against *L. monocytogenes* in combination with the bacteriocin nicin demonstrated efficacy completely eliminating bacteria in 50% of cheese samples [75]. Genetically modifying endolysins to also possess a protein having the ability to destabilise the outer membrane of Gram-negative pathogens has also demonstrated efficacy with the production of artilysins. Artilysin^®^ Art-175 is a fusion of the KZ144 endolysin and the SMAP-29 peptide having antibacterial activity against MDR strains and persister strains of *P. aeruginosa*, *A. baumannii* and *Streptococcus* [76]. Furthermore, genetically modifying species used as starter cultures in fermentation processes, e.g., *Lactococcus lactis* to code endolysins is another means of phage enzymatic biocontrol. Bacteriophage genes ply118 and ply511encoding lysins specific to *L. monocytogenes* were cloned and expressed in *Lactococcus lactis* for example [74]. As with all enzymes, phage enzymes are not heat stable making their stability in food produce a possible disadvantage (Table 3), however two new thermostable endolysins have been identified Lys68 from *Salmonella* phage phi68 and Ph2119 from bacteriophage Ph2119 [25].

### 3.4. Bacterial Resistance to Phage’s

Bacterial resistance to phages is a possibility due to mechanisms including the prevention of insertion and integration into host DNA, degradation of phage DNA, inhibiting phage replication, CRISPR/Cas and modification-restriction systems, modification of bacterial structural receptors to prevent phage binding, the formation of endospores, capsules and biofilms [71]. Indeed, bacteria can manifest resistance mechanisms preventing all 6 stages of phage infection (attachment, penetration, transcription, biosynthesis, maturation, and lysis) [53]. Bacterial phage defence mechanisms also include a toxin–antitoxin system where the bacterial genome codes for a toxin which inhibits vital biochemical reactions arresting cell growth, and an antitoxin which subsequently deactivates the toxin [45]. Importantly, bacterial phytopathogens have an additional resistance mechanism termed altruistic abortive infection (Abi) systems which initiates a cell suicide in the bacterial cell to prevent phage replication [60]. Phage resistance has been identified in the plant pathogens *Erwinia carotovora* and Pectobacterium *atrosepticum* and fish pathogens *Pseudomonas plecoglossicidae*, *Aeromonas salmonicida*, and *Streptococcus iniae* [53]. Studies to date however, have shown that phage resistance mechanisms are less effective than antibiotic resistance where less virulent species of pathogens are often produced [20]. Studies have demonstrated *P. aeruginosa* displaying resistance to phage 14/1 simultaneously become more resistant to antibiotics where *P. aeruginosa* displaying resistance to phage OMKO1 become less resistant to antibiotics [80] highlighting the varied resistance response of species to phage and antibiotic expsoure Phage TLS resistant *E. coli* displayed decreased antibiotic resistance, a pleiotrophic effect [80].

Additionally, phages can mutate and evolve to compensate for bacterial resistance mechanisms [20]. The application of phage cocktails can mitigate phage resistance and broaden the target species range. Furthermore, phage cocktails can be adapted and updated to counteract emerging pathogens and pathogen resistance mechanisms [60]. Additionally, combination cocktails including antibiotics, enzybiotics, and bacteriocins may be applied. Studies have demonstrated the efficacy of a combination of a lethal dose of kanamycin with phage SBW25′2 against *Pseudomonas fluorescens* whereas a combination of the phage with a low dose of streptomycin did not prove effective [43]. Such studies demonstrate a synergism between phages and antibiotic therapy. Phage resistance remains an area of ongoing interest as many resistance mechanisms have not been fully elucidated [45], which may hinder phage application as therapeutic and biocontrol agents going forward. It is imperative that a better understanding of phage resistance is obtained before widespread application in order to prevent a similar crisis to the current public health crisis of AMR

## 4. Industrial Application Considerations

The application of phages as biocontrol agents in foods is influenced by many factors, including the food matrix, surface area, structure, bacterial species and load, inhibitory compounds and dose of phage applied [73] (Table 2). Due to the nucleic acid content of phage’s however, temperature, water content and the presence of chemicals are also likely to impact on phage stability and activity [39]. In terms of disease treatment, the oral administration of phages is challenging as phages are pH sensitive and prone do degradation by digestive enzymes in the animal gastrointestinal tract. Encapsulation in protective coatings however, may offer a more controlled release of phages for oral administration [35]. Additionally, for some bacterial pathogens there is a lack of commercial phage options currently available. Foodborne *Clostridioides difficile* for example, remains a public health risk as this robust, spore-forming, toxin producing enteropathogen is spread via the faecal oral route and is associated with many food products. There is currently a lack of phage options for *C. difficile* biocontrol due to the technical issues of spore forming anaerobic species, where in vivo animal models are used to determine phage efficacy [38]. 

Additionally, *C. difficile* phages appear to be solely temperate in nature [57] where lytic phages are desirable as biocontrol and therapeutic agents. When designing phage or phage cocktail biocontrol options, phages coding polysaccharide depolymerase enzymes should be included to increase efficacy, broaden specificity and biofilm removal [25]. An important aspect of lysogenic phage’s is also their ability to carry and transport bacterial virulence factors including AMR genes and toxin genes [14]. The bacteriophage β carrying the tox gene encoding diphtheria toxin for example is transmitted to *Corynebacterium diphtheriae* via phage horizontal gene transfer (HGT) [34]. Shiga toxin can also be transferred by HGT amongst members of the *Enterobacteriaceae* via phage activity where Shiga toxin-producing *E. coli O157:H7* (STEC O157:H7) is a significant cause of foodborne disease [81]. *Staphylococcus* food poisoning is commonly the result of enterotoxin A produced by *Staphylococcal* species. This toxin is coded by a bacteriophage that can perform either a lysogenic or a lytic cycle [34]. Indeed, *Staphylococcal* pathogenicity islands (SaPIs) are highly mobile chromosomal islands that code for virulence factors which are transmitted via bacteriophage activity. These pathogenicity islands may harbour a set of genes for toxic shock syndrome toxin, enterotoxin B, amongst others [82]. For use in food production, it is essential to use lytic phages to avoid this HGT and potential spread of virulence factors amongst bacterial species [35]. Nevertheless, phages still offer excellent food biocontrol options as they themselves do not contain any additives or preservatives in their formulation and several are certified Kosher, Halal, and Organic, and have no impact on the organoleptic, nutritional, and rheological properties of the food [24]. Food and animal sources (slurry) may act as sources of phages which can be applied in food biocontrol, liquid manure for example was used to isolate bacteriophages showing features characteristic for the family Enterobacteriaceae which are important food pathogens [83]. Enterobacteriaceae including the important foodborne pathogen *E. coli* as described by the research of Grygorcewicz et al., (2019) relating to the dairy industry [84]. Phage biocontrol may also be amenable to Smart Farming and the application of the Internet of Things where smart technology is integrated in the farming process allowing for the detection of pathogens, weather and field conditions allowing for targeted treatment [59]. 

## 5. Conclusions

To ensure food safety, environmental safety and sustainability in food production there is an urgent need to develop green alternatives to reduce, eliminate or control pathogens in food production. Bacteriophages are a key component of all ecosystems, aquatic and terrestrial where they play a key role in bacterial evolution. The application of bacteriophages as biocontrol agents at pre-harvest, harvest and post-harvest offers many advantages for improving food safety and sustainability in line with the SDGs. Phages are potent, specific, self-replicating and organic predators of bacterial species which may be applied as therapeutic agents in disease mitigation, as disinfection agents at farm level and as bio-preservatives at food production. Importantly, phage and phage enzymes demonstrate activity against bacterial biofilms, an important aspect of bacterial virulence in food production. Phages are increasingly recognised as GRAS for application in food products and are considered organic and Kosher. Phages particularly phage cocktails have also demonstrated excellent activity against MDR species and can be used in conjunction with other safe antimicrobials such as bacteriocins to enhance activity and selectivity. Issues which must be fully investigated however, relate to phage resistance mechanisms, phage ability to transmit pathogenicity genes, phage traceability in the environment, and phage formulation and stability issues relating to therapeutic application. Indeed, phages have an innate ability to combat phage resistance mechanism in bacterial species. In agriculture and aquaculture application, a possible limitation relates to phage sensitivity to UV light and the presence of chemical contaminants. Additionally, regulatory frameworks need to be fully established detailing regulatory requirements of phages, GM phages, phage cocktails and phage antimicrobial combinations. The movement of consumers towards natural, organic, chemical free foods puts pressure on food producers to implement greener pesticide options. Phage biocontrol agents offers a sustainable means of meeting this market.

## Figures and Tables

**Table 1 antibiotics-11-01324-t001:** Bacteriophages as potential control measures for microbial food pathogens.

Food Borne Pathogen		Route of Transmission	Phage Demonstrating Efficacy
Gram negative	*Shigella* spp. *(S. flexneri, S. sonnei, S. boydii* and *S. dysenteriae)*	Soft cheese, dairy, vegetables, meat products, water, contact via fomites [20]	lytic phage Sfk20 [23] ShigaShield™ cocktail [23]
	*Acinetobacter baumannii*	Fruit, vegetables, meat, fish, dairy, water [9,10]	pIsf-AB02 via endolysin activity
	*E. coli species * (STEC, O157:H7)	Fruit, vegetables, meat, fish, dairy, water transmission	Pyo-bacteriophage, Intesti-bacteriophage, EcoShield, Ecolicide^®^ (Ecolicide PX™), Secure Shield E1 [24]
	*Pseudomonas aeruginosa*		Pyo-bacteriophage, Intesti-bacteriophage [24]
	*Salmonella* spp. *(S. enterica, S. thymiurium)*	Fruit, vegetables, seafood, [20] dairy, poultry	Salmonella typing phage 12, SJ2, SCPLX-1, SalmoFresh™ [23] SalmoPro [24]
Gram positive	*Staphylococcus species* *(Staphylococcus aureus, MRSA)*	Unwashed handled foods, meat and meat products, poultry, egg products, milk, dairy products, salads, cream-filled pastries and cakes, sandwich fillings [9,10]	vB_SauS-phi-IPLA35, vB_SauS-phi-SauS-IPLA88 [23] SES-bacteriophage, Intesti-bacteriophage [23], Stafal^®^ [25]
	*Listeria monocytogenes*	Fish and fish products, mixed meat, cheese, ready to eat food [26], pasteurized milk, ice cream, raw vegetables, raw poultry [26]	ListShield™ (formerly LMP-102), PhageGuard Listex™ (formerly Listex™; P100)
	*Enterococcus faecalis* and *Enterococcus faecium*	Meat, food of animal origin	*Podoviridae* phages—EF62phi, Orthocluster VI, phage phiFL4A [23]
	*Clostridioides difficile*	Meats, vegetables, and shellfish	myovirus ΦMMP02, φCD119 and phiCDHM1, φCD27 and ΦMMP04 and two morphologically distinct siphoviruses (SVs) φCD6356 and φCD38-2
	*Clostridioides perfringens*	Poultry meat	INT-401TM [14]
	*Campylobacter jejuni*	Raw or undercooked poultry products water [24]	Φ2, *C. jejuni* typing phage 12673, P22, 29C [23]

**Table 2 antibiotics-11-01324-t002:** Outlining the advantages and disadvantages of bacteriophages as food biocontrol agents.

Advantages	Disadvantages
Highly specific—infecting only one species of bacteria thereby unlikely to induce dysbiosis in the consumer [14]	Large scale production of phage’s and phage cocktails to meet the needs of growing food sector
Do not affect the organoleptic properties of food	Predicting which pathogen/s may be present is needed to ensure the correct phage or phage cocktail is applied
Relatively unaffected by other food preservation methods	Phage stability over the duration of food storage
High potency—small quantities required to kill bacteria	Phage resistance is unpredictable
Efficacy demonstrated against bacterial biofilms	Phages may denaturate at high temperatures
Some products currently considered GRAS [24]	Water chlorine content affects phage efficacy
Self-replicating requiring low doses [20]	The release of pro-inflammatory compounds (endotoxins and peptidoglycans) from lysed pathogens [20]
Broad application range including pre- and post-harvest	Bacteriophage-encoded toxins, e.g., botulism toxin, diphtheria toxin, cholera toxin, Shiga toxin, and pathogenicity islands [30,31]
Green technology, animal and human biocompatible	Efficacy may be affected by the food matrix [24,32]
Effective against MDR species [31,33]	Crude phage lysates may contain bacterial endotoxins [34]
Evidence of efficacy against AMR Enterococcus [35,36,37]	
May offer treatment and control of C. dificile [38]	

**Table 3 antibiotics-11-01324-t003:** Outlining the advantages and disadvantages of phage derived proteins as biocontrol agents.

Advantages	Disadvantages
No resistant bacteria evident to date [25]	Thermostability issues
Enzymes have a broader range of specificity	Large scale production issues
No risk of transferring virulence genes	Potentially inhibited by the food matrix
Penetration of biofilm matrix	Enzyme saturation kinetics
Safe for food application	May need outer membrane destabilisers present which may be toxic
Requires small quantities for Gram-positive inhibition [77]	Not self-replicating
Relatively fast action [77]	May be influenced by pH variations [76]
Can be used in conjunction with other biocontrol measures	Consumer opinion relating to GM phages and/or their enzymes
Safe for animal use, selective for prokaryotes	Shelf-life, storage issues
Effective against MDR species [76]	
Some efficacy against bacterial spores [57]

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
