# Peer review of "Bacteriophages and Food Production: Biocontrol and Bio-Preservation Options for Food Safety"

_antibiotics, 2022, doi:10.3390/antibiotics11101324_

Round 1

Reviewer 1 Report

Review of ‘Bacteriophages and food production: biocontrol and bio-preservation options for food safety’

The review provides a very useful and up to date overview on how phages can be used to improve food safety. Please see my comments below:

General comments:

·      Most sections all have one long paragraph, I strongly suggest you separate them into multiple paragraphs to make it easier to read and follow. Currently as the paragraph is so long and packed with lots of information it’s difficult to follow.

Specific comments:

Section 2

·      Line 84 – change to ‘are present’

·      Line 90 – ‘via host receptors’ section, separate this into a new sentence and expand as its not clear what you mean. I suggest you have something like ‘Phages attach to bacterial receptors…..’

·      Lines 116 to 120 – what do you mean by ‘where phages were absent in the muscle….’, its unclear. Is this in the group that received the phage treatment? So after the infection was cleared phages did not linger?

Section 3

·      Lines 192 to 196 – you state the presence of phages in the started negatively impacts fermentation and I think you should say the phages within the culture are lysing the bacteria in the starter cultures. Where do these phages come from? We did a study and found the farmers boots were full of phages and could potentially be where the phages were coming from

·      Lines 213 to 215 – the C. perfringens phage product has not been regulated and they have published one paper investigating the efficacy of the product

·      Lines 247 to 251 – does this relate to approved products or experimental studies?

·      Line 304 – ‘phage’ is repeated twice

·      Line 317 to 319 – has the product been approved?

·      Lines 474 to 477 – the sentence needs more clarity. What I understand is that one phage that become resistant to phage also became resistant to antibiotics. However another strain which become resistant to phage was still susceptible to antibiotics. Is this correct? If so I suggest re-writing the sentence to make this clear. Also are you suggesting that there is not one rule and resistance is phage or strain specific? It’s an interesting point and could be worth discussing further in the text

·      Line 483 to 485 – is ‘did not’ a typo? As the beginning of the sentence suggests the combination was successful

Comments of table:

In Table 1 in section route of transmission ensure all lists in this column begin with a capital letter as currently it’s a mix of lowercase and uppercase starting off the lists. Also what about Clostridium perfringens, it can be transmitted via meat products. I know it’s a major pathogen for the poultry industry so I suggest you include it in the table.   

Table 2 – again use capital letters at the start of the sentence. Also, some sentences have full stops whilst others don’t, pick one and be consistent.

Reviewer 2 Report

This is well-organized work that summarizes recent knowledge about phage use in food protection. 

for me, there is no information on the use of bacteriophages as herd protection against cross-contamination - e.g., by adding bacteriophages to slurry tanks in piggeries [doi: 10.3906/vet-1710-102] - thus could be an interesting approach

in the paragraph "Bacteriophages PA13076 and PC2184 at 1 × 108 PFU/mL also proved effective at inhibiting S. enteritidis in pasteurised whole milk at refrigeration temperature providing ca. 4 log loss of bacterial viability [52]. Similarly, phage LPSE1 demonstrated the ability to decontaminate milk contaminated with S. enteritidis at 28°C achieving a ca. 2.37 log bacterial death [25]." there is a lack of E. coli research in milk - you can use work: https://doi.org/10.1111/jfs.12747

Minor errors:
line10: "Goals" should be "goals"
line33: "One health" should be "One Health"

Please check all bacterial names, if bacteria is mentioned first time the full species name should be provided e.g. here: "Phage resistance has been identified in the 470 plant pathogens E. carotovora and P. atrosepticum and fish pathogens P. plecoglossicidae, A. 471 salmonicida, and Streptococcus iniae [53]. "
